# Towards modelling emergence in plant systems

Melissa Tomkins[1] [ORCID]

[1] Computational and Systems Biology, John Innes Centre, Norwich, United Kingdom

## Perspective

plant modelling; emergence; complex systems.

**Corresponding author:**
Melissa Tomkins;
Email: Melissa.Tomkins@jic.ac.uk

## Abstract

Plants are complex systems made up of many interacting components, ranging from architectural elements such as branches and roots, to entities comprising cellular processes such as metabolic pathways and gene regulatory networks. The collective behaviour of these components, along with the plant's response to the environment, give rise to the plant as a whole. Properties that result from these interactions and cannot be attributed to individual parts alone are called emergent properties, occurring at different time and spatial scales. Deepening our understanding of plant growth and development requires computational tools capable of handling a large number of interactions and a multiscale approach connecting properties across scales. There currently exist few methods able to integrate models across scales, or models capable of predicting new emergent plant properties. This perspective explores current approaches to modelling emergent behaviour in plants, with a focus on how current and future tools can handle multiscale plant systems.

## 1. Introduction

What does it mean to say that plants are complex systems? Is the number of components the sole determinant of a system's complexity? A mechanical clock, made up of many gears, cogs and springs, is a good example of a system with a large number of interacting components. The function of a clock is to keep time, carried out through the rotation of its hands, and the manner in which its components are assembled enable this designed function. To determine whether a clock is a complex system, we can consider the set of core features proposed to define complex systems, which include hierarchical organisation, feedback, nonlinearity, spontaneous order, robustness without central control, and emergence (Ladyman et al., 2013). When we evaluate clocks based on these features, it becomes clear that they would fail on most of these (with the possible exception of hierarchical organisation). Therefore, we can say that clocks are not complex systems.

Plants, on the other hand, do contain all of the core features of complex systems outlined above. They are decentralised systems that self-assemble their components across different temporal and spatial scales. There are many non-linear interactions, both within the plant and between the plant and its environment. Plants need to maintain homeostasis of water, gases, temperature and nutrients often within highly changeable environments, and this is achieved by buffering and feedback regulation. Although plants do have great phenotypic plasticity (Sultan, 2003), meaning a single genotype can result in varying phenotypes based on environmental factors such as nutrient and water availability, and stressors, many features of plant structure are remarkably consistent. Hierarchical organisation is a characteristic feature of biology, which is full of multilevel hierarchies from small molecules to macromolecules, to functional complexes, to subcellular compartments, to cells; from simple multicellular organisms to highly complex forms (Vanchurin et al., 2022). But what about the last of the core features outlined by Ladyman et al. (2013): emergence. What exactly is emergence?

Emergence describes any phenomenon of a system that cannot be predicted through the study of its individual parts, but is explainable by the collective activity of its parts (Long & Boudaoud, 2019). So, returning to the clock example, whilst it is possible that there might be other configurations of the clock components that would also make the clock hands rotate, it is unlikely that they would do so in time intervals that correspond to seconds, minutes and hours. Most configurations of the clock components will likely do nothing at all. We thus have the

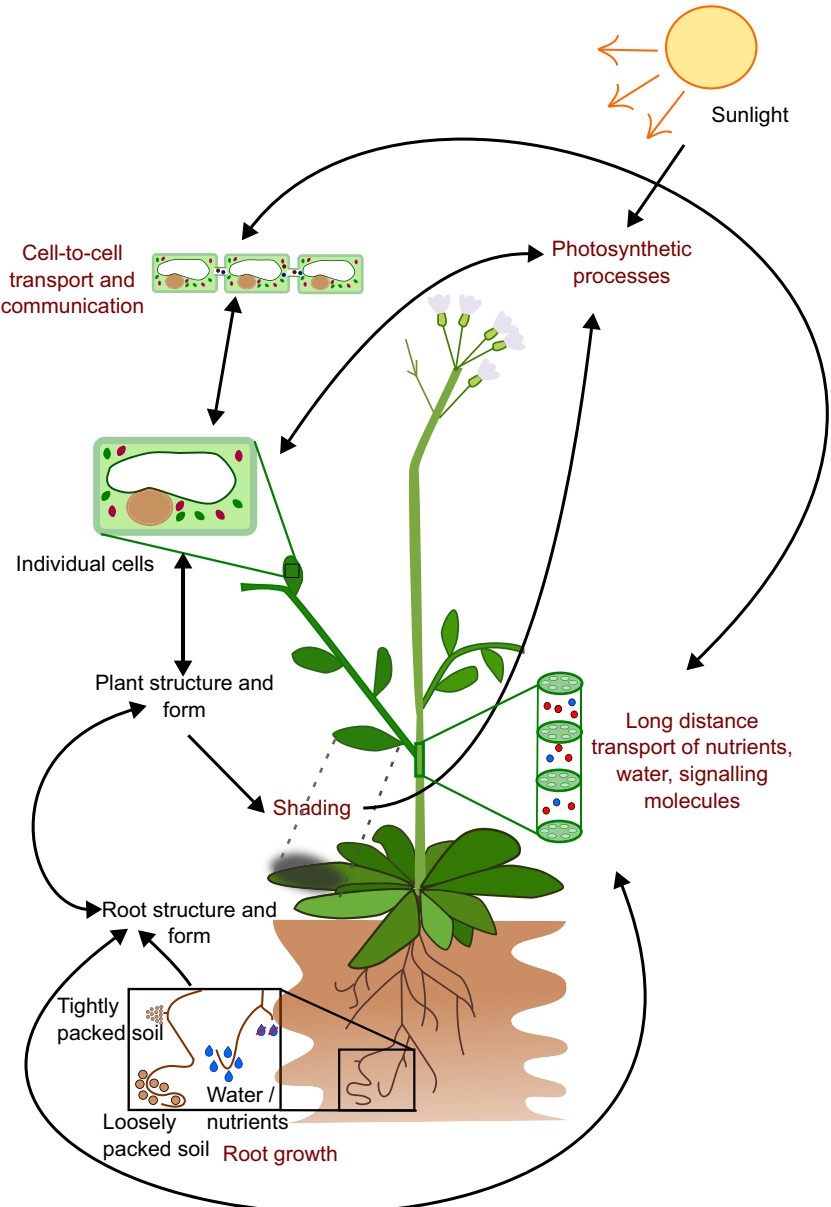

**Figure 1.** The growth and form of plants are influenced by interconnected processes that occur at different temporal and spatial scales. For instance, the growth of roots is affected by local factors such as soil composition, water and nutrient availability, as well as plant properties like the location of primordia, growth rates, and root growth angle. These interactions result in the formation of the root structure, which in turn affects the transport of water and nutrients, and shapes the overall plant structure and form, including the shape of individual cells. This structure also affects photosynthesis through factors such as the availability of chloroplasts and shading. On a cellular level, processes like gene expression and metabolism have an impact on photosynthesis and cell-to-cell communication. This limited example illustrates how the emergent properties of plants arise from underlying interactions and how these properties can further impact the interactions that gave rise to them.

case of a rather unique configuration with the functionality of a clock among many possible other configurations that do not have this property. Additionally, the function of a clock could be predicted using mechanics and kinematics if the configuration of all the components were known. Whilst the resulting position of the hands is a consequence of the clock components and their interactions, we wouldn't say that the function emerges.

As modellers, we are interested in reproducible patterns and in trying to understand how these patterns arise, often with the goal of predicting them. Whereas predicting emergent properties from their components alone is impossible, prediction from knowledge of the components and their interactions is possible, at least in principle. Doing this requires definition of the 'local rules' that underly particular system behaviours. For example, reproducing the complex formation of microtubule structure has been achieved through the use of simulations based on simple local rules defining microtubule behaviour after collisions, in which shallow contacts favour coalignment ('zippering') and steeper collisions tend to result in depolymerisation (Dixit & Cyr, 2004; Sambade et al., 2012). Many emergent patterns and structures of plant systems, such as cell size, cell arrangement, microtubule formation, gene regulatory network (GRN) dynamics, and gradients of morphogens, have been characterised and modelled by defining the local rules that underlie them (Long & Boudaoud, 2019). However, determining the local rules that result in emergent phenomena in plants is a significant undertaking for plant biologists and often requires extensive experimentation (Roeder, 2021).

One major challenge for plant modellers is how to link together models of different temporal and spatial scales, in order to obtain an integrated view of plant development and function. Many plant properties can be categorised as being emergent, from the shape and structure of each cell, which arise from interactions between factors across different scales, such as the plant DNA, and internal and external signals such as hormone gradients, light, the cytoskeleton, the plant wall, and space limitations from neighbouring cells (Kondorosi et al., 2000). The interactions between plant cells then produce new emergent properties, such as tissue and organ formation, regulation of plant development, and response to external stresses such as nutrient or water restriction. Each spatial scale of a plant is composed of emergent properties arising from the interactions of properties from a smaller scale, each of which can also be viewed as emergent properties of the level below (Figure 1). Given this, the notion of attempting to track a single genotypic perturbation through the resulting individual plant phenotype, and potentially, up to the organisation of many such plants across a landscape displaying heterogeneity in soil type and weather patterns seems nigh on impossible.

Creating integrated models with the ability to capture multiscale emergent properties can be simplified through the development of generalisable modelling formalisms. Many models involving complex systems or behaviours are designed to answer specific questions that cannot be addressed using more traditional techniques, rather than for the advancement of general theory. This was something highlighted over 20 years ago by Volker Grimm, which he termed as a need for a 'paradigmatic' rather than 'pragmatic' approach (Grimm, 1999). Such a focus of models on specific biological questions has also been proposed to be a barrier to progress in plant modelling (Louarn & Song, 2020). The main focus in this perspective is therefore on models and formalisms able to characterise general rules for emergence in plant systems, and as such, I will be limiting its scope to the mathematical formalisms and software able to advance general theory of plant emergent properties. This is not intended to be an exhaustive review of models relating to plant emergence, as that would be far beyond the scope of a single paper, but will rather highlight some recent key studies on modelling formalisms at two scales of plant systems: plant architecture and plant cell processes. In addition, I will discuss multiscale models that have successfully linked together different existing models, highlighting the strengths and weaknesses of such approaches. Finally, the discussion will examine collaborations designed to facilitate the integration of models, as well as highlighting how a recent organism-centric perspective aimed at characterising living systems could be utilised to integrate plant components across scales, in order to both characterise existing and predict new emergent plant properties.

## 2. Modelling emergent properties in plants

The selection of an appropriate modelling tool depends upon the plant system of interest, the question to be addressed, and the relevant time and spatial scales. Different modelling formalisms exist aimed at representing plants at specific scales, and here, we look at those aimed firstly at representing how plant architecture both influences, and is influenced by, plant development, and secondly, the interactions between cellular processes and the plant as a whole.

### 2.1. Plant architecture as an emergent property

The structure and function of plants, like any living system, are closely linked. Plants construct themselves through the process of morphogenesis, driven by the interactions between plant components and processes, their environment, and their genetic code (Zahadat et al., 2018). Additionally, plant growth and development take place in dynamic and often harsh environments, where plants exhibit remarkable adaptability in response to external factors such as resource competition, stress from pathogens, herbivores, and weather (Sultan, 2003). This raises questions about the balance between a plant's predetermined genetic code and its development through interactions with its surroundings. Thus, models of plant morphogenesis must account for the interplay between the physical and biological processes driving development across scales, as well as the plant's response to the environment.

Creating generalised models of the interplay between plant architecture and plant function requires formalisms able to simplify the wide range of complexity and diversity of plant shapes. Here, I only focus on those approaches best suited to generalised applications, and a more full description of the established mathematical paradigms for the formal representation of plant shape can be found in Pałubicki et al. (2019). Models that account for plant architecture and how it impacts and is impacted by plant processes such as growth and development are known as functional-structural plant models (FSPMs) (Letort et al., 2020). FSPMs represent plant architecture as interconnected plant components and can explicitly handle spatial distribution of both environmental and biological processes (Godin & Sinoquet, 2005), allowing them to incorporate feedback-loops between plant structure, function, and the environment (Bongers, 2020). FSPMs allow for questions relating to the interplay between plant development and plant environment to be addressed, and are able to handle variability between individual plant components. A complete review and history of FSPMs can be found in Louarn and Song (2020).

One major consideration when developing an FSPM is how to represent the plant architecture. A commonly used formalism that

can describe a wide range of plant features and types is an L-system. L-systems, short for Lindenmayer systems, are defined through a rewriting language able to generate complex patterns through repeated application of a set of production rules to a starting string (Lindenmayer, 1968). Each L-system consists of an alphabet of symbols, a set of production rules, and the initial string. L-systems are powerful tools for generating complex patterns that exhibit self-similarity and fractal properties, and have been used to model the growth of branches, roots and flowers (Leitner et al., 2010). Different extensions of the L-system language can allow for more dynamic integration of plant structure and function. Parametric L-systems have parameters associated with each rule, and can be updated according to a model of the plant physiological properties run in parallel. In addition, stochastic L-systems and context-sensitive L-systems choose rules based on random values and local contexts, such as developmental triggers, respectively (Green et al., 2020). Another formalism based on a rewriting language, relational growth grammars (RGGs) were proposed to address some of the limitations of L-systems, such as their inability to represent the relationships between symbols, and that an additional step is required in order to create the geometry (Hemmerling et al., 2008). RGGs are rewriting systems that are applied in parallel to graphs, with the nodes of the graphs being objects, such as plant organs, allowing for complete model information such as structure, geometry and internal state, to be accessible within a single representation (Hemmerling et al., 2008). RGGs are implemented in the modelling environment GroIMP, which was recently used to quantify the increase in photosynthetic rates for bent shoots compared with upright shoots in cut-rose production, while demonstrating that there was no impact on the quality of the harvestable flowers from the plants with bent shoots (Zhang et al., 2020).

Information about plant structure can also be encoded in a simple and general manner using multiscale tree graphs (MTGs) (Godin & Caraglio, 1998). MTGs are built upon the concept that plants can be regarded as modular organisms composed of distinct units or modules with similar characteristics. Each module within an MTG represents a specific part or component of the plant, and by capturing the similarities among these modules, the MTG provides a structured and scalable representation of plant architecture. L-systems and MTGs have been used as the basis for different dynamic plant modelling software packages able to represent emergent plant growth and development, by incorporating plant architectural models with plant system feedbacks and environmental conditions. The Virtual Laboratory (vlab / L-studio) is a plant visualisation and simulation tool based on L-systems (Federl & Prusinkiewicz, 1999; Karwowski & Prusinkiewicz, 2004; Prusinkiewicz et al., 2000), recently used to study phyllotactic patterns in flower heads (Prusinkiewicz et al., 2022; Zhang et al., 2021). MTGs have been used in software such as OpenAlea (Pradal et al., 2008; 2013; 2015) and AMAPstudio (Griffon & de Coligny, 2012) (Figure 2). One recent study using OpenAlea investigated canopy formation in grapevine, demonstrating that representing light interception and gas exchange for individual leaves, based on leaf nitrogen content and position in the canopy, more accurately reproduces the daily pattern of gas exchange for different canopy architectures of grapevines than using a single rate for the entire canopy (Prieto et al., 2020).

Whilst the software described above has focussed on the above ground parts of plants, OpenSimRoot is an example of a package designed to simulate plant root growth. Descriptions of the root simulations are held in XML files, and include information about the plant parameters, such as locations of root tips (primordia),

root growth rates, direction and plasticity, as well as environmental conditions such as nutrient and water availability, and soil type (Postma et al., 2017). Different plant species can be defined by changes to these parameters, and the overall root shape emerges from the interactions between each growing root and its environment. A similar approach, although following an object oriented design can be found in CRootBox (Schnepf et al., 2018), a C++ implementation of the MATLAB application, RootBox (Leitner et al., 2010). Whilst RootBox was based on an L-system formalism, the choice of a move to an object-oriented approach for CRootBox was motivated by both technical and conceptual considerations. An object-oriented approach allows for inclusion of code reuse and encapsulation, making the code easier to read and understand. Additionally, it facilitates connection of the CRootBox root model to shoot models, allowing for the development of an integrated plant system able to represent the complex interplays and trade-offs faced by plants during their growth and development (Schnepf et al., 2018). Integration with other models was further enabled by inclusion of Python bindings allowing coupling with other models, including soil and environmental models. CRootBox was demonstrated to successfully predict the response of a root with a known structure under different phosphate and water conditions (De Bauw et al., 2020).

Individual-based modelling (IBM), also known as agent-based modelling, is a computational approach that focuses on simulating the behaviour and interactions of individual entities, or agents, within a system. Each agent within an individual-based model has a set of rules, behaviours, and interactions with other agents and its environment. The model then simulates these behaviours and interactions over time, allowing the properties of the system to emerge. IBMs allow researchers to investigate complex systems in which variation between individuals influences the system dynamics. Such variation between individuals could be random, based on genetic mutation, arise due to differences in resources such as light or nutrients, or be based on the state of neighbours, such as hormones, or RNA expression.

A recent study developed an IBM representing wheat spikelet growth as the addition of individual units of either shoot, vegetative, inflorescence, or meristem blocks, which contained the initiation sites for both the vegetative and floral units (Backhaus et al., 2022) (Figure 3). The authors were testing a hypothesis based on their experimental data suggesting that this delayed transition could be attributed to the presence of opposing gradients of two specific genes, and the IBM was able to demonstrate that these assumptions were indeed sufficient to produce the previously unexplained lanceolate shape of developing wheat spikes. In the model, each newly formed meristem adheres to the same set of rules but can give rise to either a leaf, spikelet, or both based on the current gene profile of the two distinct classes. The individual variation observed among meristem agents is therefore influenced by the states of their neighbouring agents and the progression of time, as gene expression is influenced by the initiation of flowering. In this way, IBMs provide an excellent iterative tool for the testing and refining of hypotheses concerning emergent properties, in concert with experimental validation. For example, insight into the mechanisms behind experimental observations on how the light/gibberellin signalling pathway affects the properties of microtubules required to reorient growth was explored through the use of an IBM (Sambade et al., 2012). In addition, the emergence of plant root structure has been simulated with an IBM (Zahadat et al., 2018). Their Vascular Morphogenesis Controller algorithm, inspired by the distribution of common resources between

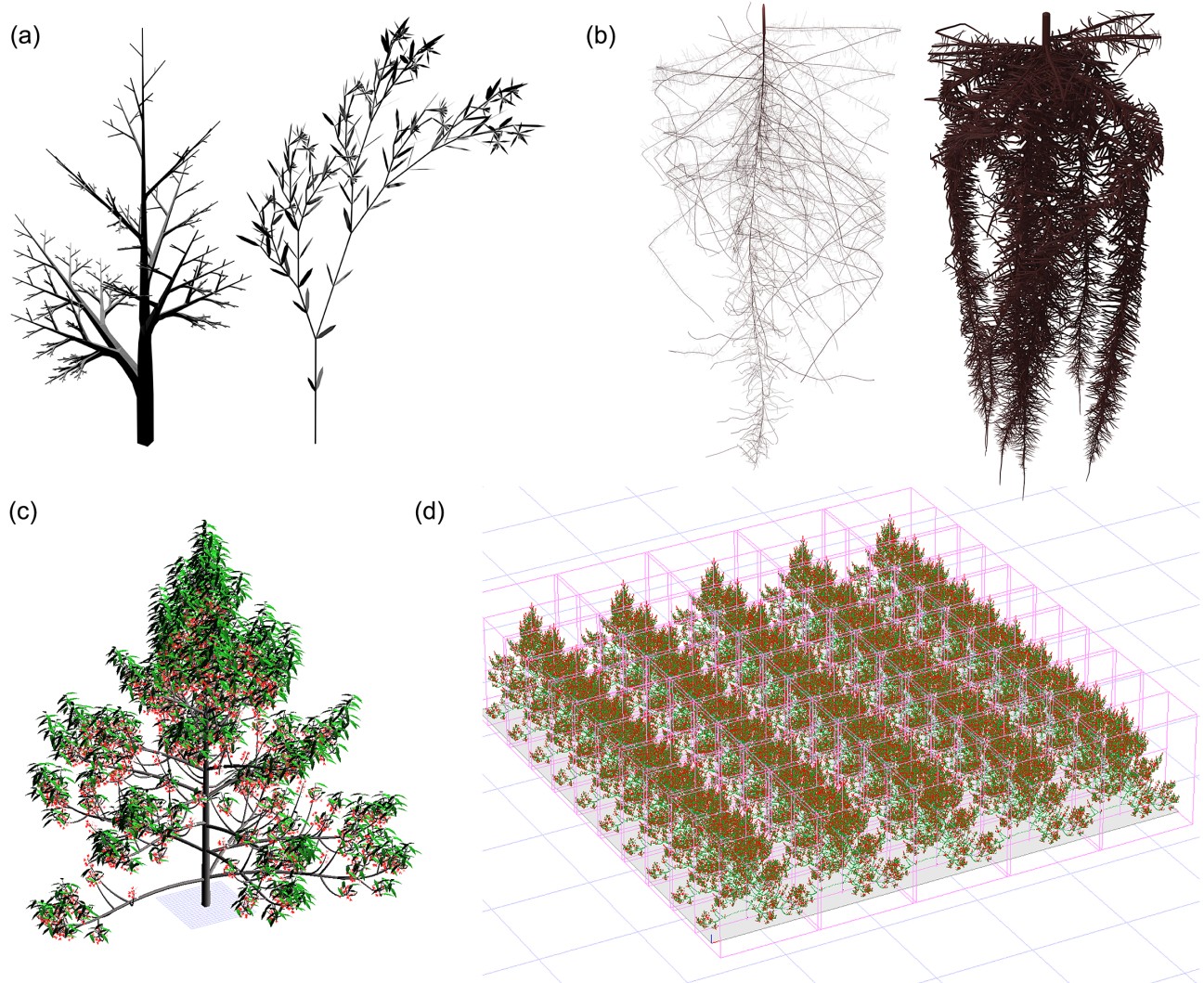

**Figure 2.** Formalisms such as L-systems and MTGs allow for the generation of complex plant shapes and fields of individual plants. (a) L-systems enable the encoding of a complex structure within simple, iterative rules, as demonstrated by this monopodial tree-like structure and plant. Tree and plant rendered in Blender, using the lsystem add-on (https://github.com/krljg/lsystem). Script for defining these systems taken from code based on Prusinkiewicz and Lindenmayer (1990) (b) Interactions between the underlying rules for the structure development and environmental conditions can allow for different structures to emerge, such as for the squash (left) and bean (right) root systems generated with OpenRootSim (Postma et al., 2017), rendered in Blender. Entire fields (d) can be generated based on a single plant (c) with AMAPstudio (Griffon & de Coligny, 2012).

the branches of plant, uses an individual-based representation of interaction between competing branches. Each branch explores its environment and produces auxin in response to light. This auxin flows towards the roots, adjusting the quality of its vessels along its way. In this way, there is positive feedback to paths that successfully transport auxin, and the system of vessel paths self-organises in a dynamic way, with nodes both being created and destroyed. Although this was a biologically inspired approach designed and tested for artificial systems, it could be applied to the development of real plant root systems, and their response to environmental stimulus, such as light and nutrients.

One current challenge for FSPMs is integrating across the whole plant. Most current 'virtual' plants focus either on the functions of the shoot or the root, and do not include the interplays between these two parts of a plant that become critical when adapting to stressful conditions (Louarn & Song, 2020). Whole plant modelling was the ambitious aim of a recent study that linked together a

reactive transport model for variably saturated media (Min3P; Mayer et al., 2012), a root architectural model (ArchiSimple; Pagès et al., 2014), and a shoot FPSM implemented in GroImp (Evers & Bastiaans, 2016), to explore the impact of soil water availability on plant development (Braghiere et al., 2020). This was a challenging undertaking, as the models were developed in different computing languages, on different platforms, and by different teams from different disciplines. Linking together models in this way requires careful handling of parameters and variables, ensuring that variables being passed between different models represent the same plant characteristic, of timescales, and also of updates in terms of whether variables are updated synchronously or asynchronously, and whether updates are frequent enough. Their model was able to successfully predict plant–plant competition and regulation on stomatal conductance to drought when parameterised under different growing conditions, demonstrating the potential for such integrated approaches for FSPMs in the future.

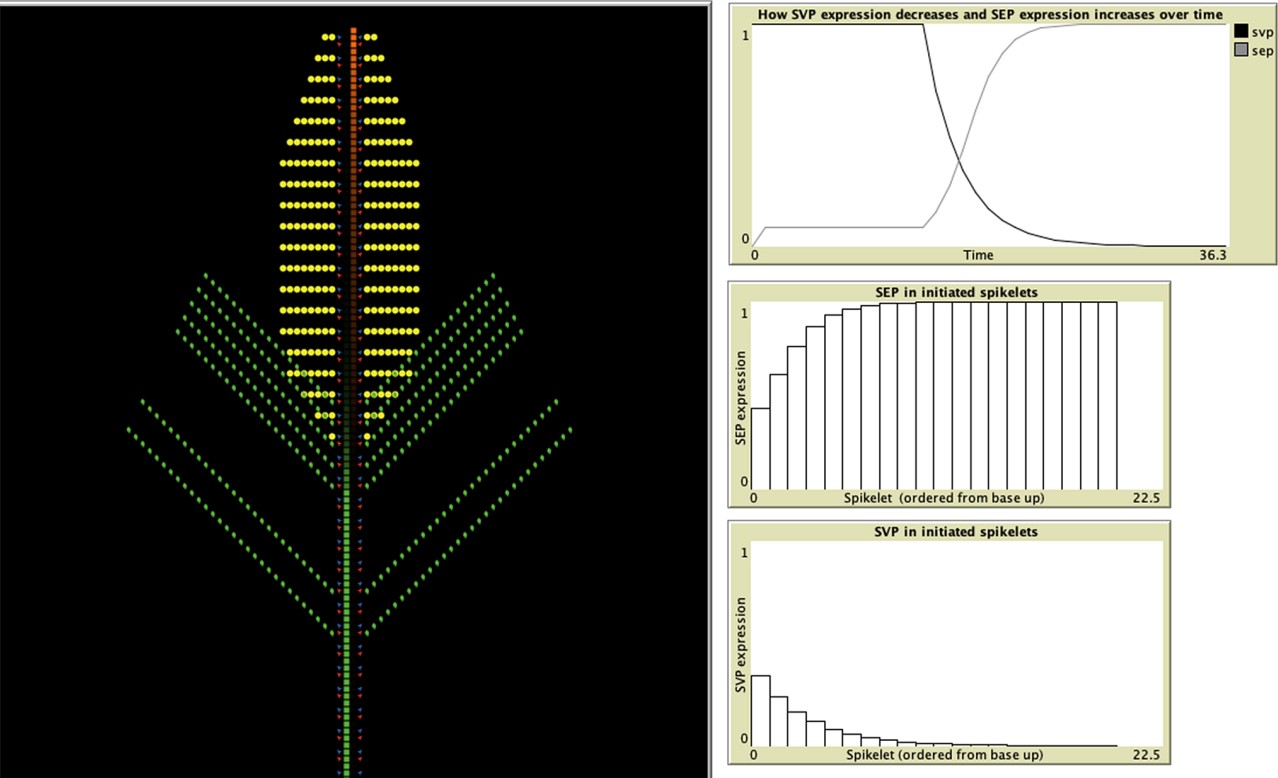

**Figure 3.** Individual-based models provide a perfect, iterative testing ground for hypotheses based on experimental data. An IBM developed in the agent-based modelling environment, Netlogo (Wilensky, 1999), demonstrates how spikelet initiation depends upon the expression of two classes of genes: SEP and SVP (Backhaus et al., 2022). The model uses expression of SEP and SVP class genes to predict when meristems (red) produce leaf tissue (green) and when they switch to producing spike tissue (yellow). SVP suppresses SEP expression, with SVP expression itself starting to decrease once flowering is triggered, allowing SEP expression to increase (top-right graph). The middle and bottom graphs depict the gradients of SEP and SVP expression, respectively, from the basal to the apical spikelets. Leaf initiation rates are suppressed by SEP, whereas spikelet initiation requires SEP. The opposing gradients of these two genes result in delayed vegetative to floral transition of the basal spikelets.

## 2.2. Emergence of cells and their functions

*"…in its complexity and functionality even the simplest, tiniest cell dwarfs everything humankind has ever been able to engineer…"*
(Wolkenhauer & Hofmeyr, 2007)

The previous section explored methods for representing the interplay between plant architecture and plant growth and development, and we now turn our attention to the emergence of the cell, and those processes resulting in its own fabrication, and its integration and response to the rest of the plant system.

Plant cells contain complex signalling networks, involving thousands of molecules, which have evolved to allow plants to respond to daily biotic and abiotic stressors (Struk et al., 2019). Defining the state of a cell requires knowledge of not only its size and shape, but also its components, intracellular reactions and interactions with the environment (Luthey-Schulten, 2021), which vary over space and time. 'Whole cell' approaches aim to represent all, or at least, the most important of these processes and interactions within an individual cell. These approaches have been proposed to have several potential benefits, such as the integration of heterogeneous datasets, prediction and understanding of multi-network phenotypes, development of new hypotheses and identification of knowledge gaps for experimental design, and the generation of frameworks for the design of genetically modified organisms (Carrera & Covert, 2015). Core cellular components, such as cellulose, starches, proteins, fats and RNA, are often

represented using particle-based reaction diffusion (PBRD). This requires implementation decisions based on the features of interest, the computational cost, and the available toolbox (Schöneberg et al., 2014). Difficulties in obtaining experimental data, combined with the challenges of simulating such systems on biologically relevant timescales necessitate decisions such as whether to use free (no boundaries) or confined particle diffusion, whether the particles are represented as points, or have specific volumes (allowing for crowding, etc.), and whether to include particle–particle interactions and potentials (Schöneberg et al., 2014). Examples of tools for cell simulations of this kind include E-Cell (Tomita et al., 1999), which has been mostly used for human and animal cells (Nishino et al., 2013; Okubo et al.,2013; Shimo et al., 2015), but could be applied to plant cells and MCell (https://github.com/mcellteam/mcell). MCell uses spatially realistic 3D cellular models and specialised Monte Carlo algorithms to simulate the movements and reactions of molecules within and between cells. MCell has recently been integrated into Blender, a free and open-source 3D computer graphics software toolset, within CellBlender (Figure 4), allowing for robust and reproducible simulations and visualisations of cell models.

While whole cell modelling has shown some success for minimal bacterial cells (Luthey-Schulten, 2021), simulating all molecules and interactions within plant cells is not computationally tractable, due to the number of particles involved. For example, it has been estimated that an average Arabidopsis

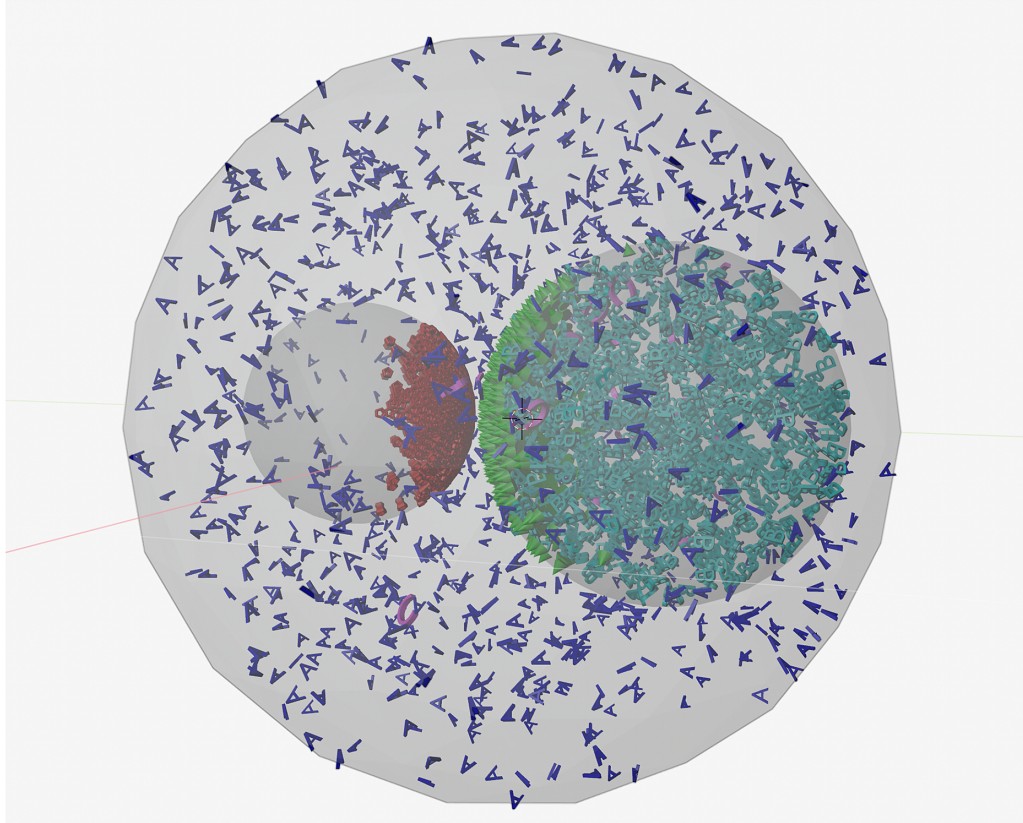

**Figure 4.** The CellBlender module for Blender can be used for the fast creation of simplified 3D cell models represent a limited number of relevant reactions. This screenshot is taken from the example model 'Organelle', and shows the interaction between surface and internal molecules of two organelles. At the start of the simulation, molecule A (dark blue) is located within the cell, outside of the organelles, and molecule B (light blue) is within organelle 2 (right). A molecules can be transported into organelle 2, through interactions with a surface molecule (green), where they interact with B molecules to produce C molecules (pink). C molecules can then interact with the surface molecule to be translocated into the cell. CellBlender development is supported by the NIGMS-funded (P41GM103712) National Center for Multiscale Modeling of Biological Systems (MMBioS).

mesophyll cell contains about 25 billion protein molecules (Heinemann et al., 2020). It has also been suggested that whole cell modelling could miss the point of mathematical modelling, which is not to realistically reproduce all molecular interactions, but to discover the general principles that determine experimental measurements (Wolkenhauer & Hofmeyr, 2007). In this regard, studying the intricate networks of molecular interactions within a cell can provide insights into the functioning of the cell as a whole. By focusing on these interconnected networks, researchers can gain a deeper understanding of the underlying principles that drive cellular functions and behaviours, rather than aiming for a comprehensive representation of every molecular detail.

Complex GRNs within a plant determine its final form and dynamic response to the external environment (Long et al., 2008). There are countless examples of how the study of specific GRNs has improved our knowledge of plant sub-systems, including salt-response (Wu et al., 2021), and the development and physiological mechanisms that regulate floral transitions (Jaeger et al., 2013; Madrid et al., 2021). GRNs aim to represent condition specific interactions of gene expression with the expression of target genes (Tripathi & Wilkins, 2021). Representing the complete inventory of gene regulatory events in a cell would require a multitude of spatially and temporally resolved GRNs, plus all their interactions and output. The Arabidopsis genome has over 30,000 loci, meaning that a complete inventory of transcriptional regulation would have over 30,000 nodes. If each gene were controlled by 10–100 TFs,

then such a network would have between $3 \cdot 10^5$ and $3 \cdot 10^6$ edges, excluding the contribution of miRNAs (Mejia-Guerra et al., 2012). Representing this number of connections is not computationally tractable, and so partial approaches or computational methods are needed to make predictions (Tripathi & Wilkins, 2021).

One method that can be used to predict emergent properties of GRNs is through investigation of random Boolean networks (RBNs) (Socolar & Kauffman, 2003). RBNs consist of a set of nodes, each of which can be in one of two states: 'on' or 'off', represented by the Boolean values 'true' or 'false'. These nodes are connected by randomly assigned links, and each node is assigned a Boolean function that determines their state in the next time step as a function of their neighbouring nodes. The Boolean functions are assigned randomly, meaning that the rules determining the behaviour of the network are unpredictable and nonlinear. As a result, RBNs can exhibit complex dynamics, including the emergence of patterns, self-organisation, and phase transitions. RBNs provide a simple yet powerful framework for studying the behaviour of complex systems, such as GRNs, and have contributed significantly to our understanding of features such as criticality (Torres-Sosa et al., 2012), robustness (Siegal & Bergman, 2002), and evolutionary capacitance (Bergman & Siegal, 2003) all of which have been proposed to be emergent properties of GRNs.

Coupling GRNs with FSPMs can allow for integration between cellular-level processes and plant phenotypic development.

Chew et al. (2014) linked four different models—a carbon dynamic model (CDM), an FSPM describing individual leaf growth and how each leaf contributes to light capture, a photothermal model (PTM) that predicted the timing of flowering based on temperature, and a photoperiodism model (PPM), which is a gene dynamic model of the circadian clock—into a multiscale mathematical model of Arabidopsis. This multilevel model was able to make multilevel predictions, from individual plant components such as leaf biomass, to the level of processes, such as the flexibility of photosynthetic control, up to entire phenotypes, such as those shown by a developmentally misregulated transgenic line. Such an approach is therefore not only able to represent changes to emergent properties resulting from changes to individual system components, but also allows for investigation into plant function and development across scales. However, linking together different models in this way is not a simple task. Even though the models were written in the same language (MATLAB), and had been developed by the same two labs, linking them together required careful consideration of decisions such as choosing a standardised time-step, and connecting variables between models—for example, a simple ratio in one model was replaced with a more complex allocation from another model. Parameterising the combined model also requires careful handling; in order to avoid overfitting, they tried to retain the original model parameters as much as possible.

## 3. Discussion

*"The purpose of a model is to capture the essence of a problem and to explore different solutions of it."* (Grimm, 1999)

The phenotype, function, and response of plants are the result of intricate interactions among cells, networks and architecture. These interactions, which occur across various temporal and spatial scales, give rise to emergent properties in a decentralised and robust complex system (Figure 5). In this perspective, I have introduced formalisms and technologies that can effectively represent these emergent properties of plants. However, the integration of plant emergent properties across scales remains a significant challenge for plant modellers, highlighting its continued importance in the field.

Connecting multiple models as modules across spatial and temporal scales is a non-trivial undertaking. Even models written in the same language, as in Chew et al. (2014), require careful consideration and handling of time units, model parameters, and differences in the way the same variables are characterised in different models. One key problem is ensuring that the data used for validation of each model are quantitatively comparable, as recalibrating parameter values for models at different scales is time-consuming and would require coordination of data acquisition between researchers from different groups, and possibly even different disciplines (Chew et al., 2014). Although packages with

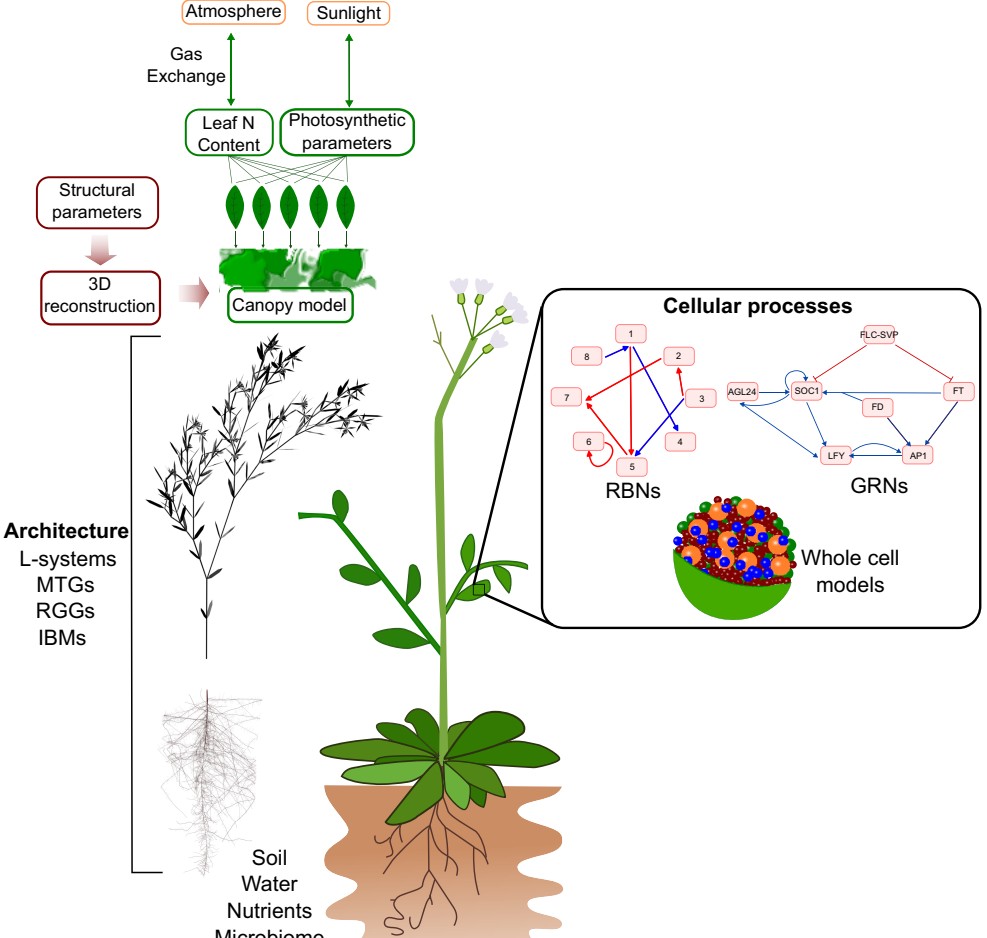

**Figure 5.** Integrated models of plant development can potentially be built using a modular approach that connects different models of underlying components, ranging from the cellular to the organ level. However, the large differences in temporal and spatial scales, underlying frameworks, and implementations complicate the linking of different modelling formalisms.

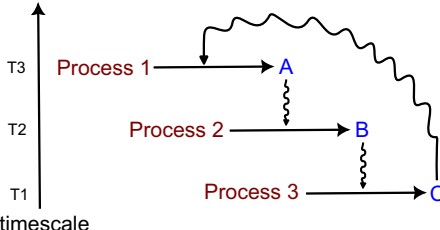

**Figure 6.** For a system to exhibit constraint closure, all of its processes (red) have to have at least one constraint (blue), and generate at least one other constraint for another process. Constraint closure allows the dynamics of the system to self-organise, and for the emergence of new properties in response to changes in conditions such as nutrient concentration, light and temperature. Figure based on Montévil and Mossio (2015).

the capability of linking together models written in different languages have been developed (Lang, 2019), differences in model formalisms and implementations are likely to complicate such integrations. Progress in modular, multiscale model development could be assisted by standardising biological computational model formulation and communication, which is the aim of the Computational Modelling in Biology Network (COMBINE) (Hucka et al., 2015). COMBINE promotes a number of standard formats for model description and analysis, such as CellML, an XML-based format for the encoding of mathematical models, Systems Biology Graphical Notation (SBGN), Systems Biology Markup Language (SBML), and BioPAX (Demir et al., 2010), a language for the representation of biological pathways.

Some researchers have proposed that the development of integrated models starts with the development of an integrated perspective. One such proposed perspective—The Theory of Organisms (ToO)—has already shown great potential in the field of animal modelling. The fundamental principles of ToO include the definition of a default cell state, which includes proliferation with variation and motility, combined with the principle of organisation by closure of constraints (Carvalho, 2022; Longo et al., 2015; Soto et al., 2016). For a model to be closed with respect to its constraints, each component must impact, and be impacted by, at least one other component (Figure 6). Constraint closure emphasises the interconnectedness and interdependencies between the various components of the model. In other words, no component within the model is isolated or independent; instead, they are linked through a network of constraints. This closure ensures that changes or perturbations in one component can propagate and affect other components, maintaining a cohesive and integrated behaviour of the model as a whole. The concept of constraint closure helps capture the complexity and dynamic nature of interactions between components, enabling a more comprehensive understanding of the system being modelled.

Models developed from a ToO perspective require the ability to represent individual differences in their components, often at the level of individual cells. Both IBMs and cellular automata have proven to be successful formalisms for building ToO models. For example, an IBM study investigating mammary ductal morphogenesis revealed the influence of mechanical forces between cells and collagen fibres, with the organisation of collagen fibres impacting cell mobility and reproductive capabilities (Montévil & Soto, 2023). Surprisingly, the model predicted occasional branching, a phenomenon observed in living organisms during mammary gland development, demonstrating the emergence of novel properties without explicit implementation of local rules. Cellular automata

have been employed to develop ToO 2D models of single-layered cell cultures. These models have explored the effects of culture geometries on tissue growth (Carvalho, 2023) and the influence of cell bioelectric properties on tumour growth (Carvalho, 2022). For instance, the latter model demonstrated how the default cell state of proliferation and motility, combined with simple rules governing bioelectric properties can shape the organisation of bioelectric properties across the cell population, ultimately determining organism size and shape.

While not yet tested extensively in plant models, ToO could be applied to well-characterised plant processes occurring across scales, such as circadian rhythms, responses to external stimuli (defence, nutrient starvation, temperature), and inflorescence timing. One of the defining principles of ToO includes motility, and while plants are sessile organisms, they still exhibit motion over a wide range of sizes and time scales (Forterre, 2013). Through processes such as the generation of turgor pressure and osmosis, plants can grow towards light, open and close stomata, and induce rapid movements in response to stimuli such as the detection of insects.

Developing models from a multilevel perspective and investigating how emergent properties interact across scales holds great potential for understanding plant development and response, it is important to remember that modelling is not an attempt to recreate realism. It might appear that a logical next step would be to work towards a complete virtual plant system, with each property emerging from its underlying model components, but such a model system seems both unachievable and undesirable. Developing multiscale models involves trade-offs between the increase in model completeness, with a corresponding increase in complexity, and loss of precision (Fish et al., 2021). Therefore, model design should include consideration of whether the addition of multiscale interactions are essential to represent the features of interest. Determining the appropriate level of resolution can be achieved through a scaling-down process, starting from a coarse model designed to explain some pattern or observation of interest. If the model is either unable to reproduce the data, or fails to capture the parameters of interest, then the model can be extended step by step on a modular basis, with checks being carried out after each extension, to ensure that it still produces the same output in a similar fashion to each previous model (Grimm, 1999). Moreover, as with any experimental design, it is important to have a clear idea of how the model data will be collected, analysed and validated. Another important factor is the scope or range of applicability of the model. Model design should seek to establish the experimental conditions that the model should cover (and with what acceptable accuracy), which conditions might be considered desirable rather than essential, and which cases are out of scope.

Finally, some of the formalisms described here involve complex models, which have their own technical considerations. For instance, individual-based models have been described as being doubly complex, as they simulate complex systems using computer code that is itself complex (Vedder et al., 2021). Structural considerations then extend from the essential system components to the software and hardware platforms on which they are run, and to how the output is stored and communicated. The many recent advances both in image analysis and sequencing technologies have paved the way for simulation methods able to incorporate complex and detailed datasets. Quantitative modelling pipelines that start from experimental data and live imaging and allow researchers to test hypotheses across scales hold great potential for generating causative links between genotype to phenotype and beyond. Whether the chosen approach is modular, or organism-centric,

understanding the interactions between emergent plant properties requires a fully integrated view of a plant system incorporating complexity across spatial and temporal scales.

## Acknowledgements

I would like to gratefully acknowledge Richard J. Morris for his numerous helpful discussions and feedback on the manuscript and Franziska Hoerbst for reading the manuscript and offering advice and corrections.

**Financial support.** This article is part of a project ('Plamorf') that has received funding from the European Research Council (ERC) under the European Union's Horizon 2020 research and innovation programme (Grant Agreement No. 810131).

**Competing interest.** The author declares no competing interests exist.

**Authorship contribution.** M.T. conceived and wrote the manuscript and prepared the figures.

**Data availability statement.** Data sharing is not applicable to this article as no new data were created or analysed in this study.

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
