## [Reviewer Report]

Submission of manuscript “Modelling of emergence in plant systems across scales” for the special collection on ‘Emergent Behaviour in Plants’

Dear Quantitative Plant Biology Editors,

I wish to submit my manuscript “Modelling of emergence in plant systems across scales” for consideration in the special collection on ‘Emergent Behaviour in Plants’ in Quantitative Plant Biology.

This review shows how an emergent perspective for plant systems across spatial and temporal scales has increased our understanding of the mechanisms behind processes including plant patterns across landscapes, plant response to pathogens, the success of invasive species, the development of plant form and function, and the interactions and networks within a cell. Furthermore, in this review I propose that a multi-scale framework integrating different complex models would constitute a valuable tool to explore how the individual model assumptions, and the interactions between the models, influence plant system level behaviour. 

Many technologies currently exist for modelling plant systems, from the ecological down to the cellular level, and this review summarises the current research, discussing the considerations behind selecting each approach. As such, it will be of great interest to modellers, and quantitative plant biologists who are interested in understanding the spatio-temporal complexity of plants and plant systems.

I think that Quantitative Plant Biology would be an ideal journal for this review that would allow it to reach its target readership most effectively. 

I thank you for your consideration and look forward to hearing from you.

Sincerely, 

Melissa Tomkins

Corresponding Author:

Melissa S. Tomkins

Department of Computational and Systems Biology 

The John Innes Centre 

Norwich Research Park 

Norwich NR4 7UH

Email: melissa.tomkins@jic.ac.uk

---

## [Reviewer Report]

While both reviewers agree that the subject of the article is of interest, they have also indicated several major points in need of improvement. One such point is what this review -which in its subject is not novel- adds to existing work, and a clearer formulation of the directions the field should move towards in the discussion. Different parts of the paper are not always clearly or logically linked, and there is a strong focus on modeling approaches with less attention for what this has brought in terms of biological insight. We ask the author to carefully consider the reviewers suggestions to amend these issues.

---

## [Reviewer Report]

Dear Melissa,

Due to unavailability of the reviewers that read the first version of your manuscript, a second set of reviewers was invited to evaluate the revised version of your manuscript. As you will see in their comments very similar issues were raised as by the first reviewers. After having a look at the manuscript myself I agree that these issues have indeed not yet been resolved.

Although we appreciate your efforts in revising the manuscript, it still appears to suffer from a lack of coherence and focus for it to deliver a strong message or conclusion, and as such it will require extensive major revisions. Both reviewers have provided concrete suggestions for such a revision. As an alternative to such extensive rewriting you may choose to submit your manuscript elsewhere.

Kind regards,

Kirsten ten Tusscher

---

## [Reviewer Report]

As you will see from the reviewers comments, despite them appreciating that the manuscript has significantly improved, they still raise some important issues. Most important among those is the extent to which matters such as models being suited for discovering and deciphering emergent properties are being explicitly highlighted in sections coming after the introduction, or the link with plant modeling to the final more philosophical and animal research oriented section.

At the same time we appreciate that you have put in considerable efforts and went through several iterations. As a solution, we propose to change the article format from review into perspective. This would target the article somewhat more as an introduction to non experts to enhance their capacity to interact with modelers and would require relatively limited rewriting.